# Cancer-Associated Fibroblasts in Cholangiocarcinoma: Current Knowledge and Possible Implications for Therapy

**DOI:** 10.3390/jcm11216498

**Published:** 2022-11-02

**Authors:** Michele Montori, Chiara Scorzoni, Maria Eva Argenziano, Daniele Balducci, Federico De Blasio, Francesco Martini, Tiziana Buono, Antonio Benedetti, Marco Marzioni, Luca Maroni

**Affiliations:** Clinic of Gastroenterology, Hepatology, and Emergency Digestive Endoscopy, Università Politecnica delle Marche, 60126 Ancona, Italy

**Keywords:** CAF, cancer-associated fibroblast, cholangiocarcinoma, desmoplasia, α-SMA, stroma, prognosis, therapy

## Abstract

Cholangiocarcinoma (CCA) is an aggressive neoplasia with an increasing incidence and mortality. It is characterized by a strong desmoplastic stroma surrounding cancer cells. Cancer-associated fibroblasts (CAFs) are the main cell type of CCA stroma and they have an important role in modulating cancer microenvironments. CAFs originate from multiple lines of cells and mainly consist of fibroblasts and alpha-smooth muscle actin (α-SMA) positive myofibroblast-like cells. The continuous cross-talking between CCA cells and desmoplastic stroma is permitted by CAF biochemical signals, which modulate a number of pathways. Stromal cell-derived factor-1 expression increases CAF recruitment to the tumor reactive stroma and influences apoptotic pathways. The Bcl-2 family protein enhances susceptibility to CAF apoptosis and PDGFRβ induces fibroblast migration and stimulates tumor lymphangiogenesis. Many factors related to CAFs may influence CCA prognosis. For instance, a better prognosis is associated with IL-33 expression and low stromal IL-6 (whose secretion is stimulated by microRNA). In contrast, a worst prognosis is given by the expression of PDGF-D, podoplanin, SDF-1, α-SMA high expression, and periostin. The maturity phenotype has a prognostic relevance too. New therapeutic strategies involving CAFs are currently under study. Promising results are obtained with anti-PlGF therapy, nintedanib (BIBF1120), navitoclax, IPI-926, resveratrol, and controlled hyperthermia.

## 1. Introduction

In the last 20 years, cholangiocarcinoma (CCA) incidence and mortality are increasing worldwide. CCA incidence is estimated at 0.3–6 cases per 100,000 inhabitants yearly in Western countries and at more than 6 cases in some regions of East Asia. CCA-related mortality is increasing as well (1–6 per 100,000 inhabitants per year, globally) and disease prognosis remains extremely poor, causing 2% of all cancer-related deaths per year worldwide. That depends on the lack of screening modalities, tumor aggressiveness, early intrahepatic or lymph node metastasization, resistance to conventional chemotherapy and limited therapeutic options [1,2,3,4,5,6].

CCA is characterized by a strong desmoplasia. The most important components of such stroma are the cancer-associated fibroblasts (CAFs), that are non-tumor cells involved in tumor microenvironment regulation. CAFs are also involved in various desmoplastic tumor such as pancreatic, lung, breast, colon, and head and neck cancers and melanoma. They play a heterogeneous role promoting immunosuppression, ECM production, and angiogenesis [7].

CAFs are objects of growing interest, since their different functions and phenotypes have been linked to cancer progression, drug resistance, and prognosis in CCA. In this review, we discuss the origin and heterogeneity of CAFs, their crosstalk with tumor and other stromal cells, their prognostic value and the potential therapeutic options associated.

## 2. Cancer-Associated Fibroblasts Origin and Phenotypic Subpopulations

Intrahepatic CCA is mainly characterized by the presence of an abundant desmoplastic reaction, consisting of a dense, fibrotic, and infiltrating stroma enriched with collagen type 1 fibers, extracellular matrix (ECM) proteins such as periostin and tenascin C. The major cellular components of this desmoplastic stroma are CAFs, with inflammatory cells, such as tissue-associated macrophages (TAMs), tumor-associated neutrophils (TANs), regulatory T lymphocytes (Tregs), and natural killer cells (NK) [8,9].

CAFs are a very heterogeneous group of cells, mainly consisting of fibroblasts and myofibroblast-like cells, that differ for origins, phenotype, morphology, and functions. It is now well established that they are not merely part of the desmoplastic stroma, but are one of the main actors determining the progression and aggressiveness of CCA. CAFs are usually identified by the presence of fibroblast’s typical phenotypic markers, that are mainly alpha smooth muscle actin (α-SMA), located in the cytoplasm [10,11], platelet-derived growth factor receptor β (PDGFRβ), in particular isoform β [12,13], and fibroblast specific protein-1 (FSP-1 or S100A4), expressed in cytoplasm and nucleuses [14]. Other markers are alpha 1 collagen type I (COL1α2), vimentin and fibroblast activation protein alpha (FAP) [15], mucin-like transmembrane glycoprotein podoplanin, and the cell surface metalloprotease cluster of differentiation 10 (CD10) [6,8,16,17,18]. Interestingly, Xuguang Yang et al. [19] studied the FAP expression in mouse liver tumors. They found that FAP was not expressed in normal fibroblasts, confirming its role as a CAF subtype marker. FAP+ CAFs had the highest expression of the CCL2 gene, but also the highest ratio of p-STAT3/STAT3 levels. FAP+ CAFs were also involved in the activation of NF-kB [20,21].

CAF origins are not fully understood yet, but they likely derive from hepatic stellate cells (HSCs) [22], periductal, or portal fibroblasts (PFs) [23], mesenchymal stem cells, pericytes, adipocytes, and circulating bone marrow-derived circulating mesenchymal cells [24,25,26]. For several years, many studies tried to demonstrate that CAFs could originate from the epithelial-to-mesenchymal transition (EMT) of cholangiocytes; however, to date, robust evidence to support this hypothesis is lacking [27,28].

In various models of intrahepatic CCA, most of the CAFs (91.9% ± 2.8%) expressed HSC markers, while a smaller portion of CAFs (6.3% ± 2.3%) showed PF markers. This proportion was then confirmed in intrahepatic CCA and hilar CCA specimens [20]. Min Zhang et al. [29] performed a single-cell RNA sequencing analysis on unselected single cells derived from human intrahepatic CCA and adjacent tissues. Specifically, analyzing CAF transcriptomic signatures, six different fibroblast subclusters were identified. The first subcluster was the most abundant, consisting of 57.6% of CAFs, and was named “vascular CAF” (vCAF) because of the presence of microvasculature signature genes and inflammatory chemokines (IL-6, CCL8). They found a relationship between vCAFs and proliferation and microvasculature metagenes. Using immunohistochemistry (IHC) staining in intrahepatic CCA samples, they found CD146+ vCAFs in the tumor core and microvascular region; this evidence implies a possible major role of vCAFs in the interaction with malignant cells.

The second cluster, the “matrix CAF” (mCAF), was characterized by the expression of high levels of typical ECM molecules, such as collagen molecules and periostin, and low levels of α-SMA. At the IHC staining evaluation, mCAFs were localized in the invasive front of intrahepatic CCA, suggesting a role in the invasiveness of CCA.

The “inflammatory CAFs” (iCAFs) expressed low levels of α-SMA too, but high levels of FBLN1, IGFI, IGFBP6, SLPI, SAA1, C3, and C7 suggest their involvement in immune modulation.

Another subcluster was characterized by the expression of MHC-II, CD74, HLA-DRA, and HLA-DRB1; such CAFs were therefore called “antigen presenting CAFs”. The “EMT CAFs (eCAFs)” expressed epithelium-specific marker genes. The last subcluster, the “lipofibroblast-c5-FABP1”, was involved in the lipid metabolism.

A similar analysis was performed by Silvia Affo et al. [20]; usingsingle-cell RNA sequencing in murine and human CAFs, different subpopulations of CAFs that partially correspond with the other previously were described. In addition to iCAF, “myofibroblastic CAFs” (myCAFs) and “mesothelial CAFs” (mesCAF), coexpressing portal fibroblast/mesothelial markers, were described. Finally, some CAFs could fit multiple subpopulations, so they called them multi-CAFs. The ones not included in one of those categories were defined as “other CAF”. iCAFs, expressing a hepatocyte growth factor and myCAFs, expressing hyaluronan synthase 2 but not type I collagen, were part of the HSC group. Both HSC-derived CAFs have shown intense interactions with tumor cells, promoting intrahepatic CCA growth via CAF subtype-specific mediators but not type I collagen.

Xiao-fang Zhang et al. [18] studied histopathological features and immunohistochemical expressions of FSP-1 and α-SMA in intrahepatic CCA to differentiate CAFs. The expression levels of such molecules, together with some morphological features, were sufficient to differentiate CAFs by maturity phenotype. On one hand, they defined “mature fibroblasts” as the slim and spindle-shaped ones, while “immature fibroblasts” were the fat and plump spindle-shaped fibroblasts with one or two nucleoli, a rough endoplasmic reticulum and a pronounced Golgi apparatus, associated with FSP-1 and α-SMA expression. All these features explain the intense secretion of ECM constituents and the copious proliferative functions. Eventually, when immature CAF constituted more than 50% of the total fibroblast of tumor stroma, that sample could be defined as an immature phenotype [18].

CAFs are not only present in CCA primary tumors, but they have been found in metastatic lymph nodes too. For a better understanding of the role of the invasiveness of CAFs in intrahepatic CCA, Rei Atono Itou et al. analyzed both macro- and micro-metastatic lymph nodes (Met-LNs), performing immunostaining for α-SMA. In micro-Met-LNs they found α-SMA-positive cells in the context of cancer cell aggregation and in micro-invasion areas. This supports the idea that CAFs in the microinvasion area could come from α-SMA-positive cells. In macro metastatic lymph nodes, they found α-SMA-positive CAFs with different morphologies around cancer cells [30].

Current knowledge shows that CAFs originate from different cells from HSCs, PFs, mesenchymal stem cells, pericytes, adipocytes, and circulating bone marrow-derived circulating mesenchymal cells. Multiple CAF subpopulations have been linked with possible features of CCA (immunomodulation, invasiveness, metabolisms, etc.). Immunopathology and immunohistochemistry have demonstrated CAF localization in lymph node metastasis and described CAF maturity phenotypes. 

## 3. Cancer-Associated Fibroblast Molecular Expression and Tumor Crosstalk

The tumor microenvironment (TME) of CCA contains multiple molecules that may help tumor progression. TME is composed of nonimmune and immune cells that can provide oncogenic or antioncogenic features [31]. Extremely high numbers of CAFs are present in the stroma and are permanently activated, resulting in a broad release of biochemical signals. In particular, transforming growth factor β1 (TGF-β1), hepatocyte growth factor (HGF), epidermal growth factor (EGF), connective tissue growth factor (CTGF), stromal cell-derived factor-1 (SDF-1), extracellular matrix (ECM) components such as periostin (POSTN), tenascin-C, fibronectin, collagen type I, osteopontin, and IL-33, and a variety of matrix metalloproteases (MMPs), such as MMP1, MMP2, MMP3, and MMP9, may be released by CAFs [21,32].

Each of these biochemical signals can have a role in the progression of CCA. For instance, cancer cells overexpress CXCR4SDF-1, which is the cognate receptor of SDF-1, strongly expressed by CAFs and not by fibroblasts in the peritumoral stroma, suggesting that SDF-1 expression may markedly increase their recruitment to the tumor reactive stroma (TRS). It has been shown in cultured HSCs that SDF-1 secretion up-regulates the anti-apoptotic protein Bcl-2 and activates ERK1/2 and PI3K/Akt pathways, favoring CCA cell survival and invasiveness [6]. In addition, SDF-1 could also promote the activation and proliferation of HSCs supporting further CAF enrichment. POSTN, which is highly expressed by CD44+ CCA cancer cells and secreted by intrahepatic CCA stem cells (ICSCs), has a strong ability to attract tumor-associated macrophages (TAMs) [33], in particular the M2 subtype that supports cancer aggressiveness. Thus, POSTN can permit CCA cell invasion through the integrin signaling pathways. The interaction between TAMs and CAFs is a topic of interest since recent studies showed that the conditioned medium from HSCs stimulates the differentiation of macrophages through the production of pro-inflammatory (IL-6) and pro-fibrotic cytokines (TGF-β) [34].

Moreover, the presence of FAP+ CAFs, but not FAP- CAFs, significantly increased the presence of myeloid-derived suppressor cell macrophages and the expression of the correlated genes and decreased the frequency of IFNγ CD8+ T cells, leading to an immunosuppression [19,20,21].

Hypoxia in HSCs and cancer cells increases the expression of placental growth factor (PlGF), a VEGF family member. This molecule activates Akt/NF-kB pathways and promotes the myofibroblast-like (highly activated α-SMA+) phenotype in CAFs stimulating the invasive growth of intrahepatic cholangiocarcinoma cells [31,35,36]. 

Consistent data suggest that PDGFRβ upon binding with PDGF-DD (produced by CCA cells upon hypoxic stimulus) induces fibroblast migration, activating the Rho GTPases, Rac1 and Cdc42, and the JNK pathway [6]. Moreover, PDGFRβ stimulates VEGF-C and VEGF-A production using fibroblasts, which promote tumor lymphangiogenesis through the expansion of the lymphatic vasculature and CCA cell intravasation that contributes to the early spread to lymph nodes and, therefore, to metastasization [27]. Interestingly, it has been shown that PDGF-BB isoform (released by CAFs and binding PDGFRβ in CCA cells) activates Hedgehog pathways protecting CCA cells from TNFα-related apoptosis inducing ligand (TRAIL)-induced apoptosis [37].

Another potential actor involved in CCA cell proliferation is heparin-binding (HB) EGF released by CAFs, which binds to the EGF receptor (EGFR) on CCA cells, resulting in the activation of signal transducers and activators of transcription 3 (STAT3), promoting tumor cell migration, motility, and invasion [6,38]. 

The TNF-like weak inducer of apoptosis (TWEAK) is expressed by macrophages in the adult liver. TWEAK is modulated by the expression of Fn14 in case of injury and repair and has a role in the development, proliferation, migration, and polarization of macrophages and CAFs in the tumor site. In CCA development, TWEAK stimulates cholangiocytes proliferation through the canonical NF-κB-induced and fibrosis-mediating HSC proliferation. In fact, Dwyer et al. hypothesized that TWEAK may be a NF-kB-driven mitogen controlling neoplastic duct and CAF proliferation. TWEAK can provoke NF-kB-driven chemotaxis-associated signaling during the steps of CCA progression demonstrating that the TWEAK/Fn14 pathway is increasingly expressed in multi-species (human–mouse) CCA [39].

Zinc finger E-box binding homeobox 1 (ZEB1) is another transcription factor expressed by CAFs in CCA cells that promotes mesenchymal transition and stemness in tumor cells. Its expression is correlated with cellular communication network factor 2 (CCN2) in myofibroblasts and CCA stroma. Thus, ZEB1 plays a key role in CCA progression by regulating tumor cell-CAF crosstalk, leading to tumor dedifferentiation and CAF activation [35].

CAF activity is guided by multiple factors and genes. A key role is played by microRNAs, which are post-transcriptional gene regulators and are involved in many cancer-related events because of their oncogene or tumor suppressor functions. In particular, Qin et al. demonstrated that a lower expression of miR-34c in exosomes isolated from CCA cell lines is associated with IL-1β, IL-8, and especially IL-6 secretion in CAFs, promoting fibroblast activation, stimulation of Wnt signaling pathways, tumor cell proliferation, migration, and invasiveness of cancer cells. This happens through the activation of cadherin [40,41]. Moreover, the high expression of miR-34c may suppress tumor growth. Thus, the downregulation of miR-34c may transform fibroblasts into CAFs through the modulation of the WNT1 pathway in CCA [41].

Another important marker expressed in several tumor cells and CAFs is podoplanin, which is a transmembrane glycoprotein and has a role in cell migration and invasion through the protrusion of the cell membrane. Podoplanin enhances the extra organ extension of the tumor and promotes tumor progression [42].

Interestingly, CAFs are not only promotors of tumor development but they may be a double-edged sword for CCA due to their characteristics. In fact, CAFs derived from HSCs have enhanced susceptibility to apoptosis. BH3-only proteins are initiators of cell death by promoting activation of the multidomain proapoptotic Bcl-2 proteins Bax and Bak. In activated CAFs, the translocation of Bax and BH3-only proteins to mitochondria leads to apoptotic cell death. This does not happen in quiescent fibroblasts and CCA cells. Moreover, antiapoptotic multidomain Bcl-2 proteins, such as Mcl-1, a resistance factor for this apoptotic pathway, is minimally expressed by CAFs or quiescent fibroblasts, although it is expressed by CCA [43]. 

## 4. Cancer-Associated Fibroblasts and Prognosis Correlation

CCA prognosis is poor due to the advanced-stage disease at the time of diagnosis and lack of effective treatment options. In fact, there is a high risk of lymphovascular metastasis, which occurs in 60–70% of patients with CCA, and only few patients undergo a curative hepatic resection. Unfortunately, the recurrence rate after surgical resection is still high and just about 5% of patients survive beyond 5 years [44]. 

The high expression of α-SMA is a negative prognostic factor because it is correlated with larger tumor size and a worse 5-year survival rate (6% vs. 29%) [45]. In particular, intrahepatic CCA is associated with the presence of lymph node metastasis and a higher histological grade, confirming that α-SMA-positive CAFs may be an important factor promoting the progression of intrahepatic CCA [46]. POSTN as well, which is expressed in α-SMA+CAFs but not in normal/cirrhotic liver or hepatocellular carcinoma, is involved as a prognostic factor in CCA because it is correlated with a shorter 5-year survival time in post-resected cholangiocarcinoma [11,47]. 

The maturity phenotype has a prognostic relevance too. Xiao-fang Zhang et al., studying stromal fibroblasts, found that FSP-1 expression (immature CAF phenotype) enhances tumor metastasis via facilitation of MMP-13 production. Moreover, intracellular FSP-1 stimulates fibroblast movement and their interaction with surrounding cancer cells. This is associated with lymph node metastasis with poorer 5-year overall survival [18].

Another important factor involved in cancer progression is the high expression of SDF-1, which is associated with tumor fibrogenesis and EMT, predicting poor prognosis in CCA patients significantly reducing median survival time [48].

It has been shown that podoplanin-positive CAFs in the tumor microenvironment confer a worse prognosis because of the higher risk of lymph node metastasis and tumor progression [42]. In fact, podoplanin may have a role in EMT, which is a possible pathway for metastasis and resistance to chemotherapy [49].

PDGF-D is involved in the prognosis because of the pro-lymphangiogenic enhancement induced by the chemotaxis of lymphatic endothelial cells (LECs) and their assembly in a proper vascular system favoring CCA cell intravasation [27].

High levels of IL-6 can promote cancer progression through epigenetic alterations and are associated with a shorter overall survival. Instead, it has been shown that high LC3 and low p62 in CCA cells is correlated with better overall survival. Tumors with low stromal IL-6 and high cancer cell LC3 expression have a better response to chemotherapy than tumors with any other combination [29,50].

It has been shown that interleukin 33 (IL-33) could be a predictive marker for good survival. In fact, in a recent study Yangngam et al. demonstrated that high levels of IL-33 in both intracellular and extracellular CCA cells suppressed cell migration. Furthermore, there is a correlation between high levels of IL-33 in both CCA cells and stromal CAFs and better 2-year survival for patients. Thus, high IL-33 in the tissues may be a biomarker for a good prognosis in CCA patients [51] (Table 1).

## 5. Cancer-Associated Fibroblasts and Potential Therapies

CCA is resistant to conventional chemotherapy and has limited therapeutic options. Actual therapies are difficult to deliver to tumor cells due to the desmoplastic and hypovascularized stroma. Thus, the depletion of this barrier through new therapeutical strategies is warranted. Here, we discuss some of the most promising molecules under investigation (Table 1).

Navitoclax (BH3-only protein mimetics) induces selective apoptotic cell death in α-SMA+CAFs compared with cholangiocarcinoma cells and quiescent fibroblast. The downregulation of Mcl-1 and the increase in Bax protein levels contributes to the sensitivity of CAFs to navitoclax-mediated apoptosis. When tested in vivo, navitoclax treatment caused a marked reduction in tumor burden and metastasis due to a decrease in lymphatic vascularization without affecting blood vascularization. In animal models, Navitoclax showed a significant improvement in survival by stimulating CAF apoptosis with a consequent reduction of α-SMA levels and the decreased expression of the pro-invasive extracellular matrix protein tenascin C [27,43].

Nintedanib (BIBF1120), a tyrosine kinase inhibitor of PDGFR, FGFR, and VEGFR, is reported to suppress the HSC activation ending in the reduction of liver fibrosis in mice. Nintedanib suppresses CAF proliferation and αSMA expression in CAFs. Moreover, nintedanib inhibits the conditioned medium of cultured CAF (CAF-CM) proliferation and the invasion promotion of intrahepatic CCA cells. Thus, combination therapy with nintedanib and gemcitabine may be a promising treatment for intrahepatic CCA [52].

In subcutaneous patient-derived xenografts in NSG mice, BLT2 antagonists combined with gemcitabine-enhanced apoptotic cells decreas tumor cell proliferation. Moreover, in the same study it was shown how gemcitabine increased the expression of stemness marker genes, which are reduced in a combined treatment with BLT2 antagonists [53].

Anti-PlGF therapy in liver cancers, including hepatocellular carcinoma and cholangiocarcinoma models, provokes benefits related to antiangiogenic/antivascular effects and reduces inflammation mediated by VEGFR. Moreover, anti-PlGF therapy decreases desmoplasia and hypoxia. Interestingly, in mice, anti-PlGF therapy combined with gemcitabine and cisplatin versus each therapy alone, shows enhanced tumor growth delay, associated with decreases in bloody ascites, and pleural effusions. In the same study, it has been demonstrated that combination therapy increased the overall survival compared with each single treatment [36].

Resveratrol, a nutraceutical, induces autophagy and abrogate IL-6 production in CAFs in vitro; as a consequence, they lose their ability to stimulate the motility of CCA cells. The enhanced CAF autophagy reduces EMT and cancer cell migration. Furthermore, resveratrol upregulates the expression of E-cadherin and suppresses N-cadherin expression in CCA cells. A reduction in N-/E-cadherin expression is known as mesenchymal-to-epithelial transition (MET) and it has been shown to reduce CCA invasiveness and metastatic potential. Thus, the role of autophagy modulators (inductors or inhibitors) is now under study as a potential promising strategy to treat different cancers [41,54].

Controlled hyperthermia could be a potential therapeutic strategy. Photothermal therapy is a minimally invasive hyperthermia treatment in which hyperthermia derives from optical energy conversion by nanoparticles (gold nanostars, carbon nanotubes, or iron oxide nanoflowers) upon near-infrared laser irradiation or alternating magnetic fields. Nicolás-Boluda et al. has demonstrated hybrid iron oxide-gold nanoparticles’ (GIONFs) preferential accumulation in CAFs. This led to a significant depletion of the αSMA+ population inducing a tumor stiffness reduction and consequent tumor regression [55].

The impaired production of IL-6 by CAFs caused by siRNA transfection shows CCA cell BAX-mediated apoptosis when treated with 5-FU. This can be explained by the two-hit model, where an additional stress, caused by 5-FU, stimulates the activated autophagy, induced by IL-6 suppression [50]. 

PDGF-D secreted by human cholangiocarcinoma cells stimulates the migration of human myofribloblastic cells in Boyden chambers. This could be suppressed by imatinib mesylate or by siRNA silencing the blockage of PDGF-D. These findings suggest that PDGF cross-talk could be a potential therapeutic target in CCA, albeit imatinib mesylate monotherapy has shown disappointing preliminary results. The EGF receptor inhibitor gefitinib and HB-EGF neutralizing antibody erase the myofibroblast contribution to tumor growth, progression, migration, and invasion in cholangiocarcinoma in mice. However, these effects by the EGF receptor inhibitor, similar to imatinib mesylate, seem to be very limited in human CCA clinical trials. The inhibition of the sonic hedgehog pathway by IPI-926, a semisynthetic derivative of the smoothened antagonist cyclopamine, has demonstrated that it reduces collagen type I deposition and α-SMA+CAF levels of desmoplastic pancreatic cancer in mice. Therefore, the use of hedgehog antagonists in cholangiocarcinoma may be a potential target of therapy [11,27,43].

## 6. Conclusions

The relationship between CAFs and tumor cells is not yet fully understood. However, their crosstalk has been demonstrated to play a key role in tumor growth and development. CAFs have various origins and they differentiate in several subpopulations. Each of them contributes to the enhancement of multiple CCA characteristics, such as lymph node invasion, chemoresistance (due to hypoxia and augmented stiffness), inflammatory environment, and modulation of immunosurveillance. A detailed comprehension of this biological landscape could have concrete implications in the clinical practice. The evaluation of CAF-related pathways in CCA tumoral tissue may prove useful for a better characterization of CCA heterogeneity. Specific CAF subtypes, with a distinguished secretory phenotype, could help in predicting CCA prognosis and response to treatment. Moreover, the modulation of the complex interplay between CCA cells and tumor stroma appears as a promising strategy also for new treatments aiming to sensibilize tumor cells to current regimens or modulate the critical pathways responsible for CCA aggressiveness. A variety of novel drugs that specifically target CAF-related pathways are actively being investigated and could help in devising new treatment options for CCA patients.

## Figures and Tables

**Table 1 jcm-11-06498-t001:** Molecules expressed by CAFs and their involvement in treatments and prognosis.

	Activated Pathways	Therapeutic Correlation	Prognostic Value
IL-33	promoting cell proliferation and extracellular matrix deposition	N/D	higher expression correlates with good prognosis
SDF-1	activation and proliferation of HSC; upregulate BCL-2, ERK-1/2, PI3K, and AKT pathways	N/D	higher expression correlates with poor prognosis
POSTN	attract TAMs, enhances CCA cell invasion	N/D	expression correlates with shorter survival
α-SMA	enhances cell growth and invasiveness	nintedanib suppresses CAF proliferation and α-SMA expression	negative prognostic factor
PDGFR-D	induces fibroblast migration and stimulates VEGF-A/C production promoting lymphangiogenesis	imatinib has disappointing preliminary results	negative prognostic factor
PlGF	promotes CAFs myofibroblast-like phenotype; activates AKT/NFkB pathway stimulating invasive growth of intrahepatic CCA	anti-PlGF therapy has antiangiogenic effect, decreases desmoplasia and hypoxia and increases overall survival	N/D
HB-EGF	activation of STAT-3 promoting tumor cells migration, motility, and invasion	gefitinib has disappointing preliminary results	N/D
BH 3-only protein	activation of Bcl-2 proteins, promoting apoptotic cell death	navitoclax induces selective apoptotic cell death in α-SMA+ CAFs, reducing tumor burden and metastasis with improved survival	N/D
IL-6	stimulation of Wnt pathway with activation of cadherin	resveratrol inhibits CAFs IL-6 production, reducing tumor invasiveness and metastatic potential	
Podoplanin	promote tumor progression and lymph node metastasis	N/D	

Molecules expressed by CAFs and their involvement in treatments and prognosis. IL-33: Interleukin 33, SDF-1: stromal cell-derived factor-1, POSTN: Periostin, α-SMA: alpha-smooth muscle actin, PDGFR-D: platelet-derived growth factor receptor, PlGF: placental growth factor, HB-EGF heparin-binding-epidermal growth factor, IL-6: Interleukin 6, HSC: hepatic stellate cells, TAMs: tumor-associated macrophages, CCA: Cholangiocarcinoma, CAF: Cancer-associated fibroblasts, PDAC Pancreatic ductal adenocarcinoma, VEGF: Vascular endothelial growth factor.

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
