# Peer review of "Cancer-Associated Fibroblasts in Cholangiocarcinoma: Current Knowledge and Possible Implications for Therapy"

_jcm, 2022, doi:10.3390/jcm11216498_

Round 1

Reviewer 1 Report

This paper describes the role of cancer associated fibroblasts in cholanciocarcinoma. Although the paper is well written and is useful to get knowledge on cancer associated fibroblasts, it includes several problems.

1) The paper is too long and redundant. It should be shortened to 80%.

2) Compared with other cancers, what are the specific characters of CAF in cholangiocarcinoma?

Author Response

Dear,

We appreciate your gentle comments and suggestions. We carefully reviewed the paper to shorten the article and to make the style more flowing. In fact, the paper body has been reduced to around 85%. To avoid the lengthening of the paper as previously requested we added a brief overview about the role of CAFs in other tumours. We think that an exhaustive dissertation about the specific characteristics and the differences between other neoplasia, even if extremely interesting, needs a deeper and detailed analysis which could render the paper less enjoyable for readers.

Best wishes,

Michele Montori

Reviewer 2 Report

The present manuscript was submitted for the special issue “From Liver Fibrosis to Carcinoma: Physiopathological Mechanisms, Diagnosis and Treatment“ in JCM. The authors review cancer associated fibroblasts in cholangiocarcinoma, the state of knowledge and potential implications for therapy.  

Comments 

  • The authors should proof-read the manuscript. There are a few spelling errors.  

Conclusion 

The manuscript is worked out very well and covers the topic of cancer associated fibroblasts very comprehensively. There is not much to criticize. Some minor spelling errors which should carefully be corrected in proof-reading. I am a surgeon and no specialist in cellular cross-talk, but especially the part of potential treatment options is interesting and important (at least for myself). In a future with individualized tumor therapies there are potential targets who can be aimed for relating to cancer associated fibroblasts.  

The manuscript offers a very detailed and comprehensive overview of cancer associated fibroblast in cholangiocarcinoma and suits perfectly within the special issue.

Author Response

Dear,

We appreciate your gentle comments and suggestions. We carefully reviewed the paper and corrected the spelling errors.

Best wishes,

Michele Montori

Round 2

Reviewer 1 Report

None